# Hear Here: Sonification as a Design Strategy for Robot Teleoperation Using Virtual Reality

Simmons, J.[a], Bown, A.[a], Bremner, P.[ab], McIntosh, V.[ac*], Mitchell, T.J.[acd]

[a]College of Arts, Technology and Environment, University of the West of England, Bristol, UK
[b]Bristol Robotics Laboratory, University of the West of England, Bristol, UK
[c]Digital Cultures Research Centre, University of the West of England, Bristol, UK
[d]Creative Technologies Lab, University of the West of England, Bristol, UK

[*]Corresponding author: Verity McIntosh; verity.mcintosh@uwe.ac.uk

## ABSTRACT

This paper introduces a novel methodology for the sonification of data, and shares the results of a usability study, putting the methodology into practice within an industrial use case. Working with partners at Sellafield nuclear facility, we explore the effectiveness of the sonification strategy in a scenario where operators in a virtual reality simulation manage a team of semi-autonomous robots. The robots use onboard sensors to map an unexplored space within a simulated nuclear facility, and the sensor data is sonified, using principles of cognitively aligned metaphor, legibility and user comfort. Operators use the auditory information in the VR scene to identify key hazards: radiation, temperature and flammable gas. They are tasked with tagging contaminated areas and protecting the robots from harmful exposure to hazards. We offer insight into features that impact user experience and system usability within our scenario. In particular we examine the importance of aligning AI robot behaviours with user expectations, and the implications of alarm fatigue when high priority sounds are repeatedly triggered. We also discuss the potential limitations of the use case, including the suitability of virtual reality environments of this nature for regular, durational use in the workplace. Overall, results of the study suggest that our distinctive approach to sonification as a design strategy for the teleoperation of robots using virtual reality is effective and easily adopted.

## KEYWORDS

sonification, sound design, virtual reality, robotics, teleoperation, nuclear decommissioning

## 1 INTRODUCTION

From underwater surveillance [15] to disaster recovery [28], immersive technologies are being increasingly used in robot teleoperation applications. Virtual Reality (VR) enables operators to phenomenologically project themselves into a remote environment in order to observe, control or embody a robot at work. Effective and safe robot teleoperation requires human operators to perceive and interpret large quantities of information. This work focuses on nuclear decommissioning, an application where operators must be aware of the physical environment (as captured by a laser scanner or RGB-D camera) as well as environmental hazards, such as radiation, temperature and flammable gas. Robot teleoperation is used for mapping and characterising unknown spaces and these environmental hazards have significant implications for both the decommissioning process and the operational health of the robots. Plotting all information graphically risks overloading the operator's visual field, diverting attention from characterising the environment (e.g., heads-up-display elements occluding elements of the visual environment). Consequently, we are motivated to explore the use of the auditory field and conveying hazard information using data sonification. The human auditory system has a high temporal resolution, wide bandwidth, and is able to localise and isolate concurrent audio streams within an audio scene [4]. These features make hearing an indispensable channel in film and gaming, and a promising medium to explore for robot teleoperation and, in particular, nuclear decommissioning. Inspired by studies showing that multisensory integration of vision with sound improves our ability to accurately process information [26], we are investigating how sound might be used to complement visual feedback for robot teleoperation in VR.

Design guidelines for sonifcations in our use case, and methods for evaluating sonification efficacy remain relatively unexplored areas of research. As a first step, we have undertaken a usability study with participants who currently work within the industry (primarily ROV operators). In this paper we present details of our user interface, our sonification design, a usability evaluation methodology and the results of our usability study. Importantly, the outcome of our data analysis is a set of design recommendations for the use of data sonification in VR.

*VAM-HRI, 2023, Stockholm, SE*
© 2023 Association for Computing Machinery.
ACM ISBN 978-1-4503-XXXX-X/18/06...$15.00
https://doi.org/10.1145/nnnnnnn.nnnnnnn

## 2 BACKGROUND

Robot teleoperation is a complex, high cognitive load task and in a nuclear decommissioning context it requires highly skilled operators to attend to multiple camera views and controls in order to carry out tasks effectively and safely. In collaboration with operators at Sellafield Ltd, we are exploring the potential advantages of VR and immersive sound for enhanced spatial reasoning and embodied interaction.

## Sonification

Sonification is defined as *"the use of non-speech audio to convey information"* [19] and has been used effectively in a wide range of applications: sensory substitution [23], medical diagnosis [32], peripheral process monitoring [31] and visual decongestion [3]. Sigrist et al have shown that sonification supported the development of rowing skills in VR [30]. Information could be more easily interpreted when it was split between visual and audio channels. Similarly, Frid et al demonstrate that task performance can be improved by splitting data presentation between visual, audio and haptic modalities [14].

Parameter Mapping Sonification [17] is one of the most widely used techniques for converting data streams to sound and involves creating manual connections between data features and auditory parameters. While this approach is simple to implement and offers great flexibility, data to sound mappings must be considered carefully as choices directly impact on the efficacy of an auditory display [33]. The most widely used auditory parameters in the sonification literature are pitch, loudness, duration, panning and tempo [9], all of which represent perceptually salient auditory parameters that can be easily controlled using simple one-to-one parameter mappings.

An ongoing challenge for sonification is the need to create auditory data representations that are easily and correctly interpreted correctly by listeners. Kramer [18] proposes that effective mappings should complement the metaphorical and affective associations of listeners. Metaphorical associations refer to mappings in which a data variable change (i.e. rise in temperature) is represented by a metaphorically related change in an auditory variable (i.e. rise in pitch) [34]. Affective associations consider the attitudes that listeners have towards given data values, i.e., trends that might be considered 'bad', such as rising global poverty and CO2 emissions, could be mapped onto auditory features that are known to increase perceived noise annoyance [7] or stress [12]. Walker et al. [33] have demonstrated that domain knowledge is an important factor in understanding how sonifications are interpreted, implying that listeners invoke auditory expectations of how particular data features should be expressed as sound. Sonifications that correlate auditory dimensions with listener expectations have been shown to be effective in a number of studies [11, 33, 34]. Ferguson and Brewster [12] refer to this correlation as *perceptual congruency* and have demonstrated that the psychoacoustic parameters roughness and noise correlate with the conceptual features danger, stress and error. Despite these efforts to support the design and evaluation of effective sonification, the majority of sonification practitioners take an intuitive, ad-hoc approach, making 'unsupported design decisions' [22, 25].

Numerous guidelines have emerged that are intended to support the sonification design process. For example, Frauenberger et al. define sonification design as "the design of functional sounds" [13] and present a set of design patterns to help novices create effective auditory displays. DeCampo later presented a Sonification Design Space Map and iterative design process interleaving stages of implementation and listening to make salient data features more easily perceptible [6]. Participatory design methods have also been used to help define more rigorous data-sound representations. Typically, this approach involves the use of practical co-design workshops in which stakeholders and experts engage in concept design and development activities. For example, Droumeva and Wakkary iterated the design of sonification prototypes for an interactive game [8] and Goudarzi et al. developed climate data sonifications with scientists and sound experts [16].

*Sonification in Human Robot Interaction.* In the context of robotics, Zahray et al. [36] and Robinson et al. [27] have used sound as a way to provide additional information about the motion of a robot arm . Both studies demonstrated that the perception of the robot's motion and capabilities are affected by the sonification design. Sonification has also been used to supplement non-verbal gestures to convey emotion using both mechanical sounds[37], and musical sonifcations [20]. Hermann et al. use sound for complex process monitoring to enable operators to establish a better understanding of system operation. Lokki et al. demonstrated that users can use sonified data to navigate a virtual environment [21]. They compared audio, visual, and audiovisual presentation of cues, finding audiovisual cues to perform best. However, it is worth noting that participants were still able to complete some of the task in the other conditions. Triantafyllidis et al. evaluated the performance of stereoscopic vision, as well as haptic and audio data feedback on a robot teleoperation task.

It is clear that data sonification is an under explored area in HRI and telerobotics, with relatively few studies across a small number of application domains. Hence, the work we present here represents an early step in understanding the utility of sonification in HRI. We are interested in understanding the utility of sonification in this novel problem domain, consequently we developed a set of sonifications for our particular use case and developed an evaluation methodology to gain this understanding and inform future sonification design.

## 3 MULTIMODAL INTERACTION DESIGN FOR ROBOT TELEOPERATION

Humans typically perceive their environments multi-modally. In the context of UX design for virtual, augmented and mixed reality interfaces in the workplace, it is important to create harmony between expectation and experience for users to comfortably adopt these interfaces and to free up cognitive bandwidth for efficient task performance.

We have developed a VR interface for data observation and robot teleoperation in a simulated nuclear decommissioning (ND) scenario, where the goal is to map the environment and characterise (label) areas of high radiation for subsequent stages of the ND process. An additional goal for users of the system is to keep the robots as safe as possible. Consequently, the robots have sensors that are able to detect radiation, temperature, and flammable gas: environmental hazards detailed as important by experts from Sellafield Ltd. It is comprised a of a graphical UI for observing robot sensor data and operating the robots, and a data sonification system for relaying data from the hazard sensors as sound.

### 3.1 Graphical User Interface Design

The VR simulation used in this work was built in Unity and comprises four main elements: an environment of digital twins of real

objects; a movement system allowing user navigation; a system for tagging radioactive objects within the environment; an interface to allow control of individual robots. The details of each component, and the reasoning for the design choices is presented here.

**Environment Design** The environment is composed of voxel objects that are initially invisible and appear when a robot sensor detects the objects. The objects were produced by scanning real-world simulacrums of ND objects using an RGB-D camera. Making pre-rendered objects appear avoids any delays that could be incurred by rendering the objects in real-time. The floor and walls are always visible, representing information known *a priori*. This setup simulates a typical ND scenario where the floor plan of the rooms is available from blueprints, but the exact composition of objects, and location of hazards is unknown.

**Movement Controls** To enable quick and easy navigation of the environment, we utilised two movement systems. The first is the teleportation system provided by SteamVR: the user holds a button on one of the VR controllers and this projects an arc with a target point where the arc collides with the floor, when the target is placed as desired the user releases the button and teleports to that location. The teleportation arc is blocked by visible objects, i.e., when an area is unexplored by robots it can be teleported into freely. The second control system utilises the D-pad on the VR controller. Clicking forward/backward jumps the user a short distance in that direction, allowing fine movement controls without inducing nausea (as continuous motion can) as well as navigation through obstacles to more easily reach a particular robot. Clicking left/right jump rotates the user 45°in that direction, allowing the user to make large rotations without having to turn their head/body large amounts, increasing comfort and reducing cable tangle issues.

**Radioactive Areas of the Environment** The main task that users must engage in is tagging of objects that are high in radiation 4. To facilitate users in completing this task two UI elements were designed: radiation markers, and radiactive object tagging. As robots move through the environment, when they detect areas of high radiation, they place a visible marker, provided there are no other markers within a threshold radius; hence, allowing the operator to identify areas that need to be investigated. A laser pointer attached to the right controller can be pointed at objects to be tagged as radioactive, and when a button is pressed the respective chunk of voxels changes to green.

**Robots** The subsidiary tasks that users must engage in are to ensure the safety of the robots, and to utilise the robots' 'real-time listening' (RTL) capability (described fully in section 3.2) to identify hazardous areas in the environment. To facilitate these we use visual robot status indications, and waypoint control of individual robots. Each robot has an outline that can be made visible to indicate a particular status, and coloured according to the status to be displayed; to increase utility the outline visibility is not blocked by intervening obstacles. A red outline is used to give a visual indication of a priority alert, an orange outline indicates RTL mode is engaged, a green outline indicates the robot is in waypoint control mode (RTL is also active in this mode). RTL and waypoint control modes are selected for a particular robot using the laser pointer, clicking once on the robot enables RTL mode (it continues to navigate autonomously), clicking the robot again enables waypoint control mode, clicking the robot again deselects it.

Waypoint control utilises the laser pointer to place waypoints on the ground which the robot navigates between. Waypoints are placed by pointing and clicking at locations on the ground, this will spawn visible waypoints, numbered to indicate the order of navigation. Waypoints not yet navigated to can be removed with a laser pointer click (LPC). On entering waypoint control mode the robot stops moving and a 'Go' button is visible directly over the robot, navigation to waypoints may be initiated by LPC of the 'Go' button. As the robot arrives at each waypoint it is removed, waypoints can be removed by the user or by an object being discovered at the waypoints location, and this results in the robot immediately re-planning its path. When a robot is deselected it will travel to its remaining waypoints and then resume its autonomous navigation.

## 3.2 Sonification Design and Implementation

The sonifications presented here seek to metaphorically express the physical dimensions of their data sources, as well as being congruous with the conditions in which our auditory scene exists. Furthermore, it is important that when multiple sonification streams occur concurrently, each stream of the resulting auditory scene can be differentiated easily by a listener [24]. With these considerations in mind, cognitive metaphor and gestalt principles played a large role in our sound design and implementation. The sonifications aim to:

- clearly communicate the value of its underlying data feature by "performing a [...] simulation of underlying physical phenomena" [10]
- cohere to existing sonic associations users are likely to possess about the hazard
- maintain a singular discernible audio stream that maintins "temporal coherence" [29], e.g., a consistent timbre and pitch, with sounds in the stream occurring continuously or in rapid succession
- blend seamlessly with other sounds emanating from robots to provide character

All audio was implemented using Audiokinetic's Wwise and integrated into the Unity environment using C# scripts. Unless stated otherwise, all sounds in our audio scene are spatialised, i.e., they are processed as if they emit from specified point sources within the environment. Artificial reverberation is also used to simulate the characteristics of the environment. These global audio features not only enhance the realism of our "cyber-physical model" [35], but aid users in locating robots and navigating the virtual environment. The following audio elements are present in the usability study: Real-Time Listening, Notifications, Priority Alerts, UI Feedback and Ambience, each described below.

**Real-Time Listening** The principal sonifications in our virtual environment comprise three parameter mappings communicating the radiation, temperature and flammable gas levels measured by each robot. This sonification is the only modality by which instantaneous hazard levels are conveyed and referred to as 'real-time listening'. RTL is the only information users have in order to perform the task of labelling highly radioactive objects. For this reason, primary focus was given to the design of these sounds, with further sonification elements designed around this sound set. The radiation sonification (example here) is an emulation of the sound produced

| Hazard | Source | Min | Low-Med | Med-High | Max |
|---|---|---|---|---|---|
| **Radiation** | Clicks | IOI ≈ 125ms | IOI ≈ 90ms | IOI ≈ 50ms | IOI ≈ 15ms
Chirps IOI ≈ 1.9s |
| **Temperature** | 220Hz Sine Wave | LFO speed = 1Hz
FM amount = 0 | ← Linear Interpolation → | | LFO speed = 10Hz
FM amount = 10 |
| **Flammable Gas** | Reverse Cymbal
Hi-Pass 800Hz | IOI ≈ 2.2s | ← Crossfade → | Staggered IOI avg. 1.1s | Staggered IOI avg. 540ms |

Table 1: Description of mappings for RTL sonifications. Where *IOI* is Inter-Onset Interval: "the time elapsed between two successive...onsets" [10] and *FM amount* is how much of oscillator 2's output is used to generate oscillator 1's output, full range: 0-100

by a Geiger counter, consisting of short, high frequency clicks. At very high levels, these transient clicks are interspersed with short chirps, designed to draw user attention (Table 1). The sonification for flammable gas was designed using phased, high-pass reverse cymbals as source samples. As can be heard here, higher gas density decreases the duration between sample onset (Table 1). The temperature sonification, as heard here, consists of a simple sine wave pattern at a fixed pitch of 220Hz. Higher temperature increases LFO speed and FM amount (Table 1). Upon trialling RTL within a VR test environment and hearing concurrent spatialised sonification streams for a team of up to four robots, it was clear that the auditory scene could quickly become congested and stressful for users. Degradation in the perceptual clarity of the streams was especially pronounced when multiple robots were simultaneously sensing the same kind of data type, as the user would hear multiple audio streams of the same mapping from multiple point sources. On this basis, we have chosen to limit live hazard data sonification to one point source at a time by requiring the user to select a single robot on which to activate RTL mode.

**Notifications** With no robots selected, RTL is turned off, but users still need to be notified when a robot encounters a hazard. Amber Case and Aaron Day list scenarios when it is appropriate to include notifications: "the message is short and simple", "information is continually changing", "the user's eyes are focused elsewhere", "the environment limits visibility" [5]. Consequently, robots emit a short notification sound, or "earcon" [10], - short snippets of RTL counterparts - when they encounter a hazard. These spatialised sounds (listenable here) communicate both hazard type and the respective robot's relative location to the user, even if visibility is obscured.

**Priority Alerts** Priority levels are useful as a safeguarding measure and combine a robot's internal "health" and external risk factors, informing users when interventions are required to increase a robot lifespan. It is intended to convey a sense of urgency without causing alarm fatigue. High priority alerts are designed to take prescience over the rest of the auditory scene. Consequently, priority sounds for each hazard were designed using unique source samples, inspired by, rather than derived from, their RTL counterparts. Whilst there is correspondence - radiation consists of clicks,

flammable gas of noise and temperature of sine tones - high priority alerts are more melodic in nature, consisting of an ascending arpeggiated sequence. As Case and Day posit, "a melody would be difficult to miss" reinforcing our "attention to such alarms rather than detracting" [5]. However, they go on: "overt melody-making could run an additional risk of extreme annoyance from overuse". For this reason arpeggiated notes are short and don't adhere to an explicit musical scale. With priority levels normalised from 0 to 100%, high priority alerts are triggered above 80% with an additional flanger effect, in the top 10% range. Combined, this effect communicates when a high priority scenario is becoming more or less serious.

With three priority sonifications and four robots, the user could hear up to twelve concurrent alerts. To reduce the likelihood of alarm fatigue, high priority alerts are synchronised when simultaneously active, and have matching rhythm and pitch, behaving like musical instruments playing in unison. This approach is inspired by Case and Day: "because we are good at picking instruments [...] out of a complex composition [...] adjusting the volume of different sounds relative to each other, and in light of their function, is important for creating a good user experience" [5]. Rather than making high priority alerts louder we used gain reductions and filtering on all other audio streams. To accompany high priority alerts, a medium priority alert system was implemented to occur when a hazard's priority level passes the threshold of 50%. These short earcons, based on the same sounds as high priority alerts, either ascend or descend, informing the user whether the priority level is rising or falling, example here.

**UI Feedback** The final sounds emitted by robots are earcons resembling tonal 'grunts', with the tonal melody dependent on the situation. These are used to acknowledge user interactions. They also occur before a high priority alert begins sounding, creating a "two-stage signal", useful for conveying that a "complex piece of information is about to be delivered" [5]. Feedback sounds for tagging, setting and removing waypoints and confirming user interactions are stereo headlocked rather than spatialised.

**Ambience** This element represents a first order ambisonic sound bed responding to the user's head movement. This increases the a sense of presence by reducing the detachment a user may feel

when interacting with robots and objects within a cyber-physical model.

## 4 USABILITY STUDY

In order to assess the usability of our system prototype, and in particular the sonifications, a usability study was conducted with domain experts as participants. Five participants took part in the study, all of whom work for Sellafield Ltd with first-hand experience of current nuclear decommissioning processes. Participant 1 (P1) worked in the remote visual inspection team, helping to develop engineering and maintenance solutions for decommissioning, whilst participants 2 to 5 (P2, P3, P4, P5) worked as remotely operated vehicle (ROV) pilots.

The sonification has been designed to communicate information about a simulated cyber-physical model, and needs to be evaluated in context for findings to be meaningful. In doing so we were also able to analyse the interplay between the various UX design elements, and their effect on usability. The study followed a think-aloud protocol, where participants voice their thought processes and observations during task performance. A video was recorded of their actions for behavioural analysis. Following the task a semi-structured interview was used to gain insight into system usability.

To enable participants to assess system usability they were tasked with identifying and labelling objects in the environment that are highly radioactive, and maintaining robot safety by navigating them out of areas that are hazardous to their health. Hazardous areas of the environment are simulated using invisible hemispheres where hazard levels increase toward the sphere centre. Autonomous robot navigation is simulated using a series of checkpoints that the robots navigate to. The scenario is kept constant across all participants.

### 4.1 Study Protocol

The study was undertaken at Sellafield's Eagle Labs in Whitehaven UK, and was divided into the following stages:

(1) **Instruction and consent:** Introducing the study purpose and itinerary and obtaining consent
(2) **Onboarding:** Tasks and system operation were explained using a video and 10 short tutorials in VR.
(3) **Main Test Scenario:** Participants completed the tasks while assessing the usability of the system.
(4) **Interview:** Gathering qualitative data about participants' experiences of the Main Test Scenario

*Data Capture.* Screen recordings were made to capture participant behaviour during both the onboarding Tutorials and main test scenario. Internal (VR) and external (voice) audio was recorded to capture participants' think-aloud dialogue. During face-to-face interviews, the following open questions were asked in a structured order, with scope to prompt further insights where relevant:

- Describe in your own words what happened in the VR experience? What did you see, hear and do?
- What do you think your role or responsibility was within the scenario?
- Do you recall any specific sounds from the experience? And, what did they mean to you?

- How useful and reliable were the alert sounds in determining what you should do and when?
- Were the real-time listening sounds useful and reliable in determining what you should do and when?
- Did the robots perform the task adequately? Did you needed to intervene? If so, why?
- Was there anything that you found particularly satisfying or successful?
- Was there anything you found particularly frustrating or unsuccessful?
- Imagine this were your full-time job. Is there anything that you would change?
- Is there anything else that you would like to share with us before we finish?

Multiple capture methods enabled the screen and interview data to be analysed independently, holistically and comparatively. The screen data allows the analysis of different UX design elements in practice and interviews capture participants' feelings about those elements in more depth. This is useful as a participant's real-time reaction to stimuli can differ from their reported impression of it: the screen capture data may go some way to moderating the effects of courtesy bias during interview responses, for example.

*4.1.1 Data Analysis.* Data were analysed using Braun and Clarke's reflexive thematic analysis (RTA) method [1, 2]. Using an inductive approach to analysis to help "generate unanticipated insights" [1], we hoped to gain valuable feedback on the usability of our prototype and use generated themes to guide our development priorities in future iterations of our HRI system. A single member of the research team transcribed the data and performed the initial codes. The remaining team members where either present at the study or read the transcript before contributing to the generation of themes. In line with the reflexive method we acknowledge the active role that researchers play in the generation of themes.

**Phase 1: Data Familiarisation** Microsoft Word's Transcribe function was used to convert think-aloud and interview dialogue into text and errors were corrected while listening to the original recordings. This provided the opportunity to become familiar with data before annotation and data coding which was undertaken using NVivo. The written account of participant actions created valuable data extracts and left a more detailed audit trail than just coding sections of video directly.

**Phase 2: Systematic Data Coding** An initial list of preparatory codes was generated and then reviewed against annotations from Phase 1, to either merge codes in the initial list or identify new codes as required. Data extracts that involved specific UX design elements were assigned a topic or sub-topic status: Control Systems (e.g. Movement, Tagging, Waypointing), AI (Autonomous Mapping, Self-Reliance), Sonification (RTL, Spatialisation, Priority Alerts (High, Medium Alerts). Codes were also created to capture participant responses to, and feelings about, these topics. Patterns in the data could then be identified.

**Phase 3: generating initial themes from coded and collated data** Preliminary themes could then to be generated based on patterns that became apparent through the coding process. Care was taken not to conflate 'topics' with 'themes': "For themes to be patterns of shared meaning underpinned by a central concept, they

must be analytic outputs, not inputs" [2]. Concept maps and coding matrices were used at this point in the analysis to explore shared experiences of the topics covered during our study.

***Phase 4: developing and reviewing themes*** This phase of analysis required a more recursive approach. Preliminary themes were checked against their respective data extracts and modified, combined or divided. As well as providing an accurate overview of participants' feelings about the system's usability, we wanted to generate some deeper insights into which elements worked well together, which didn't, and why. Every effort was made to create "clear and identifiable distinctions between themes" [1]; however, some overlap in coding was inevitable due to there being so much interplay between the different aspects of the user experience.

***Phase 5: refining, defining and naming themes*** Themes were then refined to ensure they could be written with clarity. The research team all contributed the theme definitions and content which are set out and discussed in sections 5 and 6.

# 5 RESULTS

To help situate the results, the interplay between the different UX design elements in our system and their potential impact the user experience are visualised in Figure 1.

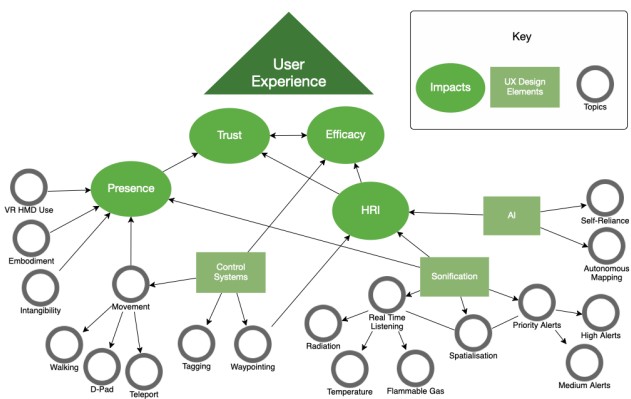

**Figure 1: The interplay of UX design elements in our system**

The occurrence of coding references can never tell the full story; however, the colour distribution in Figure 2 gives an impression of overall patterns in the data. RTL and waypointing were received particularly well by participants, with 113 combined positive responses observed against only 18 negative. All participants adopted the majority of control systems (D-pad movement, teleportation, tagging, robot selection and waypointing) very quickly during the Onboarding VR Tutorial section of the study. During the Main Test Scenario they were therefore all comfortably combining waypoints and RTL in order to investigate their environment and tag objects.

*Theme 1: Tagging Requires Sufficient Training.* Tagging had a mixed outcome, where the screen captures reveal 19 instances of participants correctly tagging radiation sources and 10 instances of incorrect tagging (Figure 2). These data extracts confirm some of the difficulty participants experienced during the tagging task:

P1: (misunderstands the tagging task, and tags an area of high temperature): *"the noise wasn't necessarily the important point,*

*but the fact it was making a noise was. Especially if you're just sort of fact finding. Everything could be highlighted, so if it was temperature or a gas leak or whatever."*

P3: (correctly identifies the high radiation object but incorrectly tags some nearby objects as well.)

P4: (incorrectly tags a high temperature object. It is apparent P4 is using the graphics and high priority alerts to complete the task and not RTL.)

P5: (incorrectly tags a high temperature object but later asks for confirmation about which hazards should be tagged and refrain from tagging any further high temperature objects, demonstrating they could then discern between radiation and temperature.)

Whilst the systems adopted easily by all participants, P1 and P4 had misconceptions about the scope of the tagging task. P3 and P5 correctly tagged multiple radiation source as well as other objects in close proximity.

*Theme 2: Sonification Is Easily Adopted.* While, many of the sonification design decisions were validated by the screen capture data, the RTL mode was discussed in more detail during the interviews:

P2: *"radiation is pretty much what you can expect to hear of radiation when you hear the measurements on site. The temperature was more of a beep and the gas was just like gas escaping. Then obviously the higher it is, the louder the radiation got; the temperature, the beeps seemed to go faster; and the gas felt like more gas was coming out. As you were leaving the area it sounded like the volume kind of depreciated as you come out of the hazardous areas."*

P3: *"it was good because you could work out straight away which items were radiated 'cause as soon as [the robot] would go anywhere near, you could just hear the radiation sound going off."*

P1: *"I was sort of guessing which [robot] it probably was by the fact that one of them would be closer to an obstacle [...] then I was using real-time listening to pick out the worst point of whatever obstacle they were moving around."*

The radiation RTL sonification considered to be clear and discernible and it is possible that participants paid closer attention to this sonification because it was an integral part of the task. However, despite finding it useful participants also noted that it was regularly interrupted by priority alerts.

*Theme 3: Better AI, Reduced Alarm Fatigue.* 14 of the 20 instances where participants expressed frustration related to the autonomous robot AI (Figure 2), with these instances being spread across 4 of the 5 participants. Furthermore, of the 21 references to AI in the data, 16 are negative feedback codes and only 3 are positive. Every time audio was spoken about in terms of frustration or distraction, it was around the topics of priority alerts and AI:

P1: *"If I was going to focus on one area, I wouldn't want [the robots] to carry on in case they get themselves in a problem state because then, I'd have to shuffle into survival mode like, 'let's go fix that'. But you wouldn't want to do that, especially if you were stuck into what could be a complex task."*

P4: *"and then other times they'd go to a high area and then they'd just stay in that area and then just start beeping *laughs*"*

P5: (ignores a high priority alert despite it suppressing RTL and the robot's red outline being visible.)

| Participant Response | Control Systems | | | Sonification | | | | | | | | AI |
|---|---|---|---|---|---|---|---|---|---|---|---|---|
| | Movement | Tagging | Waypointing | Priority Alerts | High Alerts | Medium Alerts | Real Time Listening | Radiation | Temperature | Gas | Spatialisation | |
| Affinity | 0 | 0 | 0 | 1 | 1 | 0 | 0 | 0 | 0 | 0 | 0 | 0 |
| Correct Identification | 0 | 19 | 7 | 3 | 1 | 2 | 17 | 21 | 3 | 2 | 0 | 0 |
| Easy to Learn and Use | 8 | 4 | 11 | 5 | 4 | 0 | 8 | 5 | 4 | 0 | 1 | 0 |
| Shows Understanding | 0 | 6 | 3 | 1 | 0 | 0 | 8 | 3 | 3 | 1 | 1 | 1 |
| Useful | 0 | 4 | 6 | 7 | 1 | 3 | 8 | 3 | 0 | 0 | 4 | 2 |
| Confusion | 1 | 4 | 1 | 5 | 1 | 2 | 5 | 1 | 3 | 1 | 4 | 0 |
| Discomfort | 6 | 0 | 0 | 0 | 0 | 0 | 0 | 0 | 0 | 0 | 0 | 0 |
| Distraction | 0 | 1 | 0 | 5 | 0 | 0 | 0 | 0 | 0 | 0 | 0 | 6 |
| Frustration | 0 | 0 | 1 | 5 | 0 | 0 | 0 | 0 | 0 | 0 | 0 | 14 |
| Misconception | 0 | 10 | 0 | 4 | 3 | 0 | 2 | 2 | 2 | 0 | 0 | 0 |

**Figure 2: A Coding Matrix of Participant Responses to UX Design Elements**

P1: *"Barry and Clive need to be quiet when I'm focusing on Steve."*
P5: *"the alerts were quite good...as long as there weren't too many going off at once with alarms 'cause if you have that many different robots, you'd be focused on one trying to see if it's getting louder or quieter, and if an alarm would go off it would just kinda throw you off a little bit."*

Looking at P5's actions during the screen captures and their interview responses, it is evident that when they realise that the robots in this simulation will keep returning to the most hazardous areas regardless of intervention, they give up safeguarding. Alongside other participants' responses, this shows that the lack of AI quickly led to alarm fatigue during the Main Test Scenario due to the frequency of alerts interrupting the tagging task.

*Theme 4: Exploration Requires a Map.* Our data confirms the need for the inclusion of visual markers to be placed at sites where there are high hazard levels. There are many data extracts showing participants hunting around for a radiation marker, and only then being able to utilise the waypoint/RTL/tagging functions:

P1: (Radiation markers initially prompted the use of waypoints to explore the area in more detail. They then use RTL to correctly identify a highly irradiated object).
P3: (Uses waypoint functionality after seeing a radiation marker to instruct a robot to explore an area in more detail.)
P5: (Uses combination of radiation markers, waypoints and RTL to correctly tag a highly irradiated object.)
P4: *"... it was successful that [a radiation marker] pinpoints which areas are high 'cause then you can go back to that and then try and work out if it's an object or if it's just a high area without there being an object there."*

*Theme 5: Spatialisation: A Visualisation Supplementation.* Throughout the screen captures, there are a total of 15 instances where a participant alters gaze direction in response to a robot making a sound outside their line of sight. In 100% of these instances, participants turned and looked in the correct direction first time:

P2: (A medium priority alert is triggered out of view. P2 turns their gaze in the correct direction towards the point source and watches the robot. Its high priority alert then triggers.)
P4: (A high priority alert is triggered by a robot out of view. P4 turns their gaze in the correct direction towards the point source to see the red highlighted robot.)

P5: (A hazard alert is triggered by a robot out of view. P5 turns their gaze in the correct direction towards the point source before continuing with their task.

These observations show that audio spatialisation offered participants valuable non-visual information about robots' locations whilst their gaze was directed elsewhere. Then, once fixing gaze on the respective robot, participants would also see an omni-visible red outline, highlighting the robot in a high priority state. They would then often keep this visual aid in view whilst heading towards it. The red outline was also integral in confirming which robot was alerting in cases where there was more than one robot in close proximity.

*Theme 6: The Viability of VR.* The final theme reflected participants' experiences whilst in VR and their views on using immersive technology in the workplace. This theme covers a range of participant experiences and suggestions:

P1: *"from what I'm seeing there [the usability study environment], it's probably going to be successful without any major sort of detours."*
P1: *"It is pretty intuitive to be honest. These are the things we've been speaking about since we started fiddling with robots and not just ROV's. It's almost like, 'this is the next step' isn't it? This is 'look at what you could possibly do?'... So you've put a picture of what we were trying to explain to people for a while, which is quite interesting to see."*
P4: *"It's handy in the sense that you're not getting exposed to it"*
P5: (commenting on the prospect of working full time in VR) *"I'd maybe rotate with someone and take breaks every now and then because it'd be disorientating. I'd probably prefer sitting down as well, if you were working on it. But yeah. Just the amount of time actually doing it with the headset on."*
P1: (jokingly describes the ability to navigate through walls using the D-pad as ' *'weird'*' and says they prefer teleporting.
P1: (commenting on sounds that they recall) *"there was a lot of clicking and beeping, but, knowing that you're not in any sort of risk, for me it's sort of..."*
P5: During 'Tutorial 2: Locomotion', light-heartedly says that they're *"scared to walk too far".*

The last data extract was reflected in all participants' comments on the ability to walk around their physical space whilst in VR. Participants did not use this feature, suggesting it may be redundant in future solutions.

## 6 DISCUSSION

Our results show participants easily adopting D-pad movement, teleportation, waypointing and tagging to aid exploration and HRI. This suggests that both the D-pad and laser pointer movement offered by VR controllers can be readily accepted by users when interacting with a cyber-physical model. In most instances, objects were tagged correctly and errors were either due to misunderstandings about the scope of the tagging task, or the liberal tagging of objects near the radiation source in surrounding areas of medium or low radiation. To address these issues, here are three solutions we intend to implement:

(1) Rigorous onboarding to ensure users fully understand the role of tagging in the decommissioning process
(2) Tagging for all three hazard types, encouraging a greater level of attention be paid in differentiating the hazard sonifications
(3) By enabling users to listen to hazards from their position in the environment we hope to improve the speed and accuracy of the tagging task while keeping robots safe.

Radiation sonification in particular was noted to be clear and discernible. This could be because it was the only RTL sonification required to complete the prescribed task in our usability study. However, additional factors could be that the sound was effective in the auditory scene and is based on the well established Geiger counter sonification. We intend to explore this further in a subsequent user study employing Solution 2. The implementation of Solution 3 will also give users more control over how they listen to their environment will further improve the efficacy of all our RTL sonifications.

Shortcomings with the robot's autonomous navigation (or AI) forced participants to manually safeguard the robot team and regularly interrupted task performance. Audio was simply the modality by which this interruption manifested. The robot team's lack of self-reliance was highlighted by the priority alert sonifications, which led to alarm fatigue, distraction, frustration and a loss of trust. Enhanced autonomous navigation would avoid know areas of high radioactivity and thus reduce the frequency of alerts and improve task performance. Solution 3 would also reduce alarm fatigue, as users would be less reliant on robots for accomplishing the tagging tasks and reducing the need to put them in danger.

In our system, sound is a live feed to the data but does not offer a historic record of previously mapped hazards. Due to the autonomous nature of the robot team's exploration and mapping, users will not always be present when a hazard is sensed. Consequently, visual markers are required to keep historic logs of hazard data, pinpointing areas that need to be explored by the user at a convenient time. This was confirmed by the data, as participants would always seek out a radiation marker before activating a robot's RTL mode to investigate the area's hazard levels. Spatialisation was shown to be a valuable feature of our sonification, facilitating participants' localisation of robots. Furthermore, during high priority situations, omni-visible robot highlighting also proved to be useful, removing any ambiguity about their location. This was especially helpful when multiple robots were clustered, behind walls or otherwise obscured from visibility. For this reason, we intend to use both audio spatialisation and omni-visible visual aids in future

iterations of the system. Although cyber-physical models promise better safety and efficiency in facilitating the nuclear decommissioning process, discomfort, disassociation and detachment remain potential issues, as exemplified by some of our results. Two participants expressed a need for regular breaks from using VR in the workplace, and the ability for freely walk around the environment seemed redundant. There is also the feeling of disassociation and detachment users could potentially experience when physically removed from the consequences of their interactions. Long-term research into the viability of cyber-physical models in the workplace is therefore needed.

## 7 CONCLUSION

In this paper we have introduced a distinctive approach to sonification, drawing on cognitive metaphor and gestalt theory as core design principles, and creating a sense of perceptual congruence for users that allows them to associate groups of sounds with the originating data source. With this approach, data interpretation is effectively spread across sensory modalities, freeing up cognitive capacity for users to attend to complex tasks in a hazardous, teleoperated robotic scenario using VR. The benefit of spatialised sonification has been confirmed, as 100% of participants could reliably localise the source of alerts. The efficacy of this pairing of sonification with VR can also be inferred from participants ability to attend to, interpret and act in response to the sonifications, and complete the task. Conversely, we identified issues of discomfort and disassociation that call into question the suitability of VR environments for durational working, and signal the need for further study prior to the adoption of such methods in the workplace.

The design approach to sonification appears to have been effective across a range of behavioural and self-report markers, and we hope that this model will offer a useful and replicable methodology for future researchers and sound designers interested in sharing data across sensory channels as a means to lessen cognitive load. The sounds chosen were evidently well recognised and utilised by participants, being interpreted in tandem with complementary visual information such as the robot avatars and hazard markers. Sonification data appears to have been used both in isolation, and to complement visual information when confirming and adding detail to the characterisation of a space. The use of sonification to support management of semi-autonomous robot teams, and to characterise potentially contaminated space also appears to have been validated as a viable HRI use case. Early indications suggest that design decisions taken to create affinity with the robots and thus build trust in the system have been broadly effective, however further study is required, with a wider range of approaches tested before sonification, affinity and trust can be reliably affiliated.

## CONFLICT OF INTEREST

The authors have no competing interests to declare.

## ACKNOWLEDGMENTS

This research has been made possible thanks to funding from the UKRI Trustworthy Autonomous Systems Hub (EP/V00784X/1). Our thanks to Melissa Willis and the Robotics and Artificial Intelligence

Interpreted Research Team at Sellafield, and to Steven Wood and the whole team at Digital Catapult.

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
