# OpenReview forum: "Hear Here: Sonification as a Design Strategy for Robot Teleoperation Using Virtual Reality"
_humanrobotinteraction.org/HRI/2023/Workshop/VAM-HRI — VAM-HRI 2023 Oral_

### Official Review · Program_Chairs · 2023-02-25
**Accept**

**Rating:** 8
**Confidence:** 5

**Review:**

Reviewer 1

This paper investigates the usage of sonification to improve robot teleoperation using VR devices. I recommend this paper be accepted, it is interesting, relevant to the community, and investigates a problem not yet well addressed (the use of sound in VAM-HRI).

Feedback:
- It would be helpful to have some photos of the GUI and environment.
- There should be a baseline method of communicating the parameters (radiation, temperature and gas levels) without using audio (e.g: graphically) to understand how much sonification helps compared to other ways of communicating the same information.


--------------------------------------------

Reviewer 2

This work examines how sonification can be integrated into VAM-HRI interfaces (namely for robot teleoperation) to enhance operator/supervisor effectiveness and/or situational awareness. In the context of a nuclear facility during decommissioning scenarios multiple robots are teleoperated to identify and tag hazardous areas (e.g., radioactive, flammable gases, high temperatures). A strong motivation and even stronger background section (which provides a great deal of information regarding sonification) is presented to readers of this paper, followed by a detailed and well thought out design section. The use of professionals for the evaluation of the interfaces followed by interviews (and coding and analysis) further strengthens this paper to help provide design guidance and lessons learned for the VAM-HRI community. A quote that stuck with my while reading the paper is: “with this approach, data interpretation is effectively spread across sensory modalities, freeing up cognitive capacity for users to attend to complex tasks in a hazardous, teleoperated robotic scenario using VR” which I strongly agree with as many VAM-HRI interfaces front load the sensory burden on the visual senses. I think this is a strong paper that is highly relevant to the VAM-HRI workshop, and I recommend its acceptance.

Questions and Comments:
- As mentioned above the related work section is very strong and presents the reader with a great background on sonification.
- I was pleasantly surprised to be presented with links to the actual audio sounds used within the interface which helped me better understand their descriptions in the text. This greatly strengthened the paper and helped captivate me as a reader.
- The biggest weakness of this paper is the lack of any figures that showed how the interface actually worked. I would have expected multiple figures that showcased all the different functions and operator viewpoints that are described in the paper.
- The results section was thorough and I enjoyed reading quotes form the users as well as seeing the analysis performed from the coded data.
- I think the explanation of the experiment task could be further fleshed out, at times I was left unsure as to what the exact procedure was for participants during the evaluation.
- I think Section 3.2 could user another pass for clarity in terms of how and when these sounds are used in the interface. For example, I was unsure when priority alerts were triggered (every time a hazard is encountered? A threshold?).

---

### Decision · Program_Chairs · 2023-03-02

Accept (Oral)